# Healthcare built environment and behavioural and physiological indicators of stress responses in autism spectrum disorder: Protocol for a mixed-methods systematic review

Billie Weaver[1]*, Evangelia Chrysikou[1,2], Mar García-Rodríguez[3], Daryia Palityka[1], Eva Hernandez-Garcia[1]*

1 The Bartlett School of Sustainable Construction, University College London, London, United Kingdom, 2 Clinic of Social and Family Medicine, Department of Social Medicine, University of Crete, Heraklion, Greece, 3 Faculty of Biological Sciences, University of Valencia (UV), Valencia, Spain

* billie.weaver.23@ucl.ac.uk (BW); eva.hernandez.20@ucl.ac.uk (EHG)

## Abstract

### Introduction

Autistic populations are more likely to need healthcare (HC) services due to co-occurring mental health issues, including anxiety and attention-deficit hyperactivity disorders. One of the most significant barriers to delivering optimal medical procedures in autism spectrum disorder (ASD) is the sensory overload in HC settings. Divergent sensory processing, along with unpredictable built environments (BEs), can exacerbate stress-induced anxiety and avoidant behaviour. A growing body of systematic studies links autism-friendly BEs with positive care experiences, yet substantial gaps remain in understanding the effects on behavioural and physiological aspects of emotional responses. This systematic review aims to comprehensively evaluate the HC-BE features that impact on behavioural indicators and non-invasive biomarkers of stress, anxiety and sensory processing in patients with ASD, to establish best practices.

### Methods and analyses

The Preferred Reporting Items for Systematic Reviews and Meta-Analyses guidelines will be followed. Peer-reviewed articles in PubMed, Embase, Web of Science, Scopus, and PsycINFO databases and Google Scholar will be searched. Studies will be selected if they apply qualitative, quantitative, or mixed-methods designs. Two independent reviewers will select studies at the title and abstract, and full-text screening stages. Data will be extracted by one reviewer and verified by review members using a crowdsourcing approach for quality assurance. Risk of bias will be assessed by one reviewer using the Cochrane Risk of Bias Tools, The Critical Appraisal Skills Programme, and The Mixed Methods Appraisal Tool, and checked by the reviewer with

**Data availability statement:** No datasets were generated or analysed during the current study. All relevant data from this study will be made available upon study completion.

**Funding:** The author(s) received no specific funding for this work.

**Competing interests:** The authors have declared that no competing interests exist.

methodological expertise. A results-based convergent synthesis design is planned for data synthesis.

## Discussion

This review, which converges indicators and patient experiences, will provide a complete overarching picture of the inherent complexities associated with HC-BE and autistic individuals. The findings can inform decisions and recommendations for research and practice.

## Trial registration

PROSPERO CRD42024562288

---

## 1. Introduction

Autism spectrum disorder (ASD) is a neurodevelopmental condition characterised by impairments in social communication and interaction, as well as restricted, repetitive patterns of behaviours, or activities [1]. Globally, the prevalence of ASD has increased significantly over time, with about 1 in 100 children diagnosed [2]. While autism has long been perceived as a paediatric condition by the general population, many autistic individuals do not receive a diagnosis until adulthood, especially females [3]. Meta-analyses indicate that co-occurring mental health conditions, including developmental coordination disorder, attention-deficit hyperactivity disorder, sleep-wake problems and anxiety disorders, are highly prevalent in autistic children and adults [4,5]. Approximately 35% of autistic individuals are diagnosed with an anxiety disorder; [5] however, many experience elevated anxiety symptoms that may not meet formal diagnostic thresholds [6], as well as sensory processing differences [7]. These conditions often lead to frequent engagement with healthcare (HC) services and the need for in-patient mental health unit care [8].

While the HC built environment (HC-BE) may have limited impact those who self-identify as healthy [9], it can affect vulnerable individuals and influence patients' perceptions and sense of agency, and this might not be an exception for autistic people as explored in this review. This effect may occur across intimate (i.e., small, enclosed spaces with heightened vulnerability), private, and public HC-BE settings [10]. Evidence demonstrates that autistic individuals frequently experience greater challenges across various HC-BEs—including hospitals, paediatric centres, and primary HC settings—compared to the neurotypical population [11–13]. Incorporating accessibility features and customised design elements has been shown to significantly enhance the well-being and overall experience of individuals with various developmental disorders, including autism [14]. It is well-recognised that the most significant barrier to accessing optimal HC service is the sensory overload of the environment [15]. Many autistic individuals experience sensory symptoms (up to 84%) regardless of age [16], and the HC-BE, including its spatial design, can exacerbate the overload, affecting the HC delivery and leading to stress and avoidant

behaviour [17]. Divergent sensory processing, along with unpredictable environments, can further amplify stress-induced anxiety [18]. The design of sensory adaptive environments—integrating elements such as dim lighting, and spaces for pressure touch or movement during medical procedures—has been proposed as a potential long-term solution to meet the sensory needs of the autistic population [19].

While there is a growing body of systematic reviews linking spatial criteria for the design of autism-friendly BEs with self-, caregiver- and HC professionals-reported positive experiences, a tremendous gap remains in what is known regarding the effects on the behavioural and physiological measures of feelings [18,20]. Although more research is needed, investigations using electroencephalography (EEG), electrocardiography (ECG) and functional magnetic resonance imaging (fMRI) can provide a deeper insight into how interior BE design features—such as geometry, materiality, texture, and style of furnishing—impact neural activity and structures of emotional processing [21]. Overall, autistic individuals have abnormal latency, amplitude and power of EEG and magnetoencephalography (MEG), leading to atypical sensory experiences [22]. Interestingly, altered physiological arousal evaluated through salivary cortisol (CT) has been reported in neurogenetic syndromes linked to autistic phenotypes, though associations with behavioural indices of anxiety were syndrome-specific and not consistently observed [23]. Multi-level neuroimmune markers offer complementary clinical and pathologic information in relation to BE exposure [24]. Taken together, it is crucial to adopt a multi-method approach that integrates self-reported experiences, behavioural, and multimodal biomarker-based measures to capture the complex responses to anxiety and stress in autism [25].

The evidence on which specific components of the HC-BE most effectively improve the care experience for autistic people is not well established, as the considered previous outcome measures for its assessment may have provided a limited comprehensive understanding. To the best of our knowledge, no systematic reviews have evaluated and synthesised both observational and interventional research in this field using a multi-method approach. This systematic review aims to explore whether exposure to specific HC-BE features can alter behavioural and physiological indicators of anxiety, stress, and emotion in ASD, which is associated with atypical sensory processing, by integrating both qualitative and quantitative evidence to identify effective design strategies.

## 2. Methods

### 2.1. Protocol

This protocol for systematic review and meta-analysis was conducted according to the Preferred reporting items for systematic review and meta-analysis protocols (PRISMA-P) statement [26]. The reporting of results will adhere to the PRISMA 2020 guidelines [27]. The review protocol is registered in the International Prospective Register of Systematic Reviews (PROSPERO) database (identification No. CRD42024562288). The PRISMA-P checklist is available in S1 Appendix.

### 2.2. Eligibility criteria

A summary of the participants, interventions, exposures, comparators, outcomes and study design considered according to PI/ECOS strategy (Participants, Intervention/Exposure, Comparison, Outcome, Study design), as well as the type of setting included, is provided in Table 1.

**2.2.1. Participants.** This systematic review will include (a) children and adults who meet the diagnostic criteria for ASD in the Diagnostic and Statistical Manual of Mental Disorders, 5th Edition (DSM-5) or the International Classification of Diseases, 11th Revision (ICD-11), with or without comorbidities, and with no exclusions related to age, sex, race, ethnicity, culture or educational status; (b) parents or primary caregivers of individuals with ASD; and (c) HC professionals (general practitioners, nurses, psychiatrists, doctors, psychologists, therapists, etc.) with experience in providing care to individuals with ASD. Studies reporting on mixed populations will be included if their results are reported separately, enabling extraction of data about these specific populations.

**Table 1. Summary of inclusion criteria.**

| Components | Inclusion Criteria |
|---|---|
| Participants | 1. Children and adults diagnosed with ASD according to DSM-5 or ICD-11.<br>2. Primary caregivers of autistic individuals.<br>3. HC professionals with experience in providing care to autistic patients. |
| Intervention, Exposure | 1. Interventions that modified any HC-BE features.<br>2. Exposure to any aspect of the HC-BE pre-defined in the protocol |
| Comparison | All eligible papers will be included regardless of whether they have a control group. |
| Outcomes | Behavioural and physiological indicators related to stress/anxiety and sensory processing categorised as:<br>1. Physiological biomarkers (non-invasive).<br>2. Behavioural measures (validated tools):<br>  • PROMs – generic or condition specific.<br>  • PREMs – focused on the functional aspects.<br>3. Experiences/perspectives (self-, caregiver-, or HC professionals-reported). |
| Study design | RCTs, uncontrolled trials, non-randomised study designs of interventions and exposures, qualitative studies and mixed methods studies. |
| Physical setting | Both inpatient and outpatient HC-BE. |

ASD, Autism Spectrum Disorder; DSM-5, Diagnostic and Statistical Manual of Mental Disorders, Fifth Edition; HC-BE, Healthcare-Built Environment; ICD-11, International Classification of Diseases, 11th Revision; PREMs, Patient-Reported Experience Measures; PROMs, Patient-Reported Outcome Measures; RCTs, Randomised Controlled Trials.

Throughout this review, identity-first language will be used (e.g., autistic individual) to refer to individuals diagnosed with ASD. This is based on research that has highlighted a preference within the autism community in English-speaking countries for not separating autism from the person (e.g., "individual with autism") [28].

**2.2.2. Interventions and exposures.** HC-BE is conceptualised as the physical environment of HC settings, including the ambient, architectural, landscape and interior design features, and their maintenance and housekeeping (S2 Appendix) [29]. Both inpatient and outpatient HC-BE will be included, however, those purposively designed for autistic users will not be considered. The decision to exclude care environments purposely designed for autistic users is grounded in the conceptual aims of this review. Since most autistic individuals receive care in standard HC facilities rather than specialist units, our focus is on understanding their real-world experiences within care environments that were not designed with autism in mind. To our best knowledge, national and international HC design guidelines do not include autism specific accessibility requirements, and this gap is reflected across design guidance reviewed in other research projects across multiple countries. Consequently, most HC facilities—including general hospitals, maternity services, dental clinics, primary care, and most psychiatric wards—are not purpose-built for autistic users. Including specialist autism designed settings would therefore introduce a fundamentally different design context and fall outside the scope of identifying BE features that affect autistic individuals in standard HC settings they encounter across their lifespan.

Evidence will be sought regarding the effect of (1) exposure to any aspect of the HC-BE pre-defined in the protocol, and (2) interventions that modified any HC-BE features, on the autistic individuals' behavioural and/or physiological response, even if this was not the primary focus of the research.

**2.2.3. Comparison.** All eligible papers will be included regardless of whether they have a control or comparison group.

**2.2.4. Outcomes.** A multi-method approach is adopted, including experiences and behavioural and/or physiological measures of stress and anxiety linked to sensory processing in autistic people. To this purpose, systematic reviews aiming to document validated behavioural indices and non-invasive biomarkers in human studies of ASD related to anxiety/stress

or sensory processing were identified. Multiple reviews addressed physiological biomarkers [22,30,31] and self- and caregiver-reported, behavioural tools [32,33]. After a comprehensive reading and assessment independently conducted by two reviewers (BW and EHG), the outcomes were categorised as:

1. Non-invasive biomarkers. While no biomarkers have been formally qualified by the Center for Biologics Evaluation and Research Biomarker Quality Program, attention was given to those showing the most consistent results in predicting or measuring physiological responses to anxiety, stress and sensory processing. Grouped by system, ten biomarkers were included [22,30,31]:

   - Neurophysiology: EEG, MEG – Measure brain activity patterns, including event-related potentials and oscillatory activity, associated with sensory processing and emotional regulation. EEG and MEG both measure neural activity arising from synchronous postsynaptic currents, with EEG detecting electrical potentials at the scalp and MEG detecting the associated magnetic fields; both methods can capture spontaneous and evoked activity.

   - Neuroimaging: fMRI – Evaluates brain activation and connectivity; outcomes may include changes in blood oxygenation level-dependent signals relative to baseline or control conditions.

   - Autonomic nervous system: Heart rate variability (HRV) and electrodermal activity markers (EDA) – HRV quantifies parasympathetic and sympathetic activity through variations in time intervals between heartbeats, while EDA measures sympathetic arousal via skin conductance.

   - Hypothalamus-pituitary-adrenal axis: Salivary CT – A widely recognised biomarker of physiological stress in clinical practice.

   - Peripheral immune system: White blood cell counts (WBCs), monocytes (MTs) and interleukins (ILs) – WBCs indicates changes in the immune cell composition and function induced by stress, whereas MTs and ILs primarily reflect inflammatory signalling responses. These biomarkers will be included whether measured in blood, urine, or saliva samples.

   - Behavioural: Eye-tracking markers (ETMs) – Track gaze patterns, fixation duration, and attention allocation in response to environmental stimuli.

For each biomarker, data on baseline values, post-exposure/intervention measurements, and changes over time, including diurnal variations, will be extracted whenever reported. Given the heterogeneity in biomarker collection and measurement methods, differences in sample types and modalities will be considered. Circulating biomarkers may be measured in blood, saliva, or urine (CT, WBCs, MTs, ILs), with cut-offs, normative ranges, or thresholds recorded to enable comparison across studies. Neurophysiological and neuroimaging biomarkers (EEG, MEG, fMRI) will be based on signal patterns, oscillatory activity, or blood oxygenation level-dependent responses. Similarly, autonomic measures (HRV, EDA) will be extracted according to the specific metrics reported in each study. All ten non-invasive biomarkers were included based on their consistent results in prior systematic reviews, ensuring a rigorous and unbiased selection.

2. Behavioural measures (validated instruments). Both patient-reported outcome measures (PROMs), either generic or condition specific, and patient-reported experience measures (PREMs) capturing autistic individuals' experiences and perceptions of care, including attention to physical support needs and environmental aspects (e.g., comfortable care environments) were considered. Specifically, the following instruments were included [32,33]:

   - Stress-specific questionnaires: Adjusted Stress Survey Schedule (SSS), Perceived Stress Scale (PSS), Stress in Children questionnaire (SiC), Chronic Stress Questionnaire for Children and Adolescents (CSQ-CA), and Anxiety Dimensional Observation Schedule (Anx-DOS).

- Combined questionnaires: Depression Anxiety Stress Scale (DASS), Anxiety, Depression and Mood Scale (ADAMS), Social Responsiveness Scale-Second Edition (SRS-2), and Repetitive Behaviour Questionnaire (RBQ).

- Moment-Specific "State-Like" Questionnaires: Subjective Units of Distress Survey (SUDS) and Event Sampling Method (ESM).

Although no listed, other standardised measures will be considered for inclusion.

3. Self-, caregiver-, or HC professionals-reported experiences and perspectives. Non-standardised tools (questionnaires, surveys that included yes-or-no, multiple-choice, and open-ended questions) will also be considered, as well as data from interviews, focused on experiences and perspectives of any feature of HC-BE specified in the eligibility criteria related to stress and anxiety.

**2.2.5. Types of studies.** A broad spectrum of study designs will be eligible, including randomised controlled trials (RCTs), uncontrolled trials, and non-randomised study designs of interventions and exposures, and qualitative studies. Due to the limited available research, a range of mixed methods studies that examine both objective and subjective indicators will be included. Opinion pieces, commentaries, single case studies, case reports, systematic studies with or without meta-analysis or meta-synthesis, and studies that are not peer-reviewed will be excluded.

## 2.3. Search strategy

PubMed-Medline, Embase, Web of Science, Scopus, PsycINFO and Google Scholar will be searched using a combination of controlled vocabulary terms (i.e., Medical Subject Headings [MeSH] descriptors) and keywords, with Boolean operators and truncation, that relate the three main concepts: "autism", and "HC-BE", and "sensory processing/anxiety/stress". The search strategy for PubMed will be adapted as necessary for the other databases (S3 Appendix). Searches will be limited to peer-reviewed journal articles published from January 1, 2014, until the year 2025. This date restriction was chosen because studies on the topic of interest before 2014 were not identified after a preliminary search. The validated search query filters for Humans will be added as the Cochrane Handbook for Systematic Reviews of Interventions recommends [34]. No language restrictions will be applied in the searches, and various approaches will be implemented to address any eligible non-English articles during the screening and data management processes. The database searches will be supplemented using the snowballing method, in which the reference lists of identified relevant articles are manually checked for additional relevant research.

## 2.4. Study selection

The references will be managed in Mendeley (Elsevier, London, UK). Duplicates will be automatically removed through The Systematic Review Assistant-Deduplication Module and exported to Rayyan, a web-based systematic review programme, for screening.

Titles and abstracts will be independently screened by two reviewers (BW and MGR) based on eligibility criteria. Second, the full texts of the articles will be then obtained, and the same two reviewers will independently undertake screening. To manage potential non-English articles, translation tools will initially be used to assess the eligibility of abstracts [35]. For those deemed eligible for full-text review, translation support will be requested via Cochrane Task Exchange, an online platform connecting researchers with the necessary language skills [36]. The authors will record and report any studies that do not meet the inclusion criteria. In case of disagreement, a third reviewer (EHG) will step in to resolve it. Adhering to PRISMA guidelines, a flow diagram will be generated to visualise the selection process.

## 2.5. Data extraction

Data extraction will be conducted by one author (BW) and subsequently verified by review members using the crowdsourcing approach (MGR, DP). The same members, together with those identified through Cochrane Task Exchange,

will assist the first author in handling English and non-English language articles during the data extraction, risk of bias assessment and data synthesis processes. A purpose-designed data extraction form will be piloted a priori via calibration exercises to ensure consistency. Data will be retrieved on the following variables (see S4 Appendix for further information): (a) study information: first author, year of publication, geographical location, study focus; (b) research design and methodology: duration, follow-up, data collection, statistical analysis; (c) participant characteristics: sample size, eligibility criteria; (d) clinical characteristics: comorbidities (including anxiety disorder), sensory processing characteristics (where reported), including receipt of therapy targeting sensory processing challenges, prescribed medications; (e) HC-BE: study setting, HC typology; (f) HC-BE intervention/exposure: devices, technology and/or BE feature; (g) environmental factor (if applicable): parameter type within categories, exposure levels, equipment; (h) physiological biomarkers (e.g., laboratory assay methods, cut-offs, midpoints); (i) behavioural scores (e.g., PREMs, PROMs); (j) outcomes: analysis models, effect estimates; and (k) key findings.

## 2.6. Risk of bias and quality assessment

The risk of bias assessment and quality of evidence will be completed by one reviewer (BW) and verified by the review member with methodological expertise (EHG).

For RCTs, the risk of bias will be assessed with the Cochrane risk of bias tool (RoB 2), across several features of trial design, management, and reporting. Judgement in RoB 2 is assigned as "Low", "High" or "Some concerns" risk of bias [37]. The Risk of Bias In Non-randomized Studies of Exposure (ROBINS-E) tool will provide a structured approach to assessing bias in quantitative, observational studies [38]. The ROBINS-E evaluates seven domains: confounding, measurement of the exposure, selection of participants, post-exposure interventions, missing data, and selection of reported data. Each domain is rated as "Low", "Moderate", "Serious", or "Critical" and the overall study risk of bias is determined by the highest rating across the domains.

To assess those non-randomized studies that compare the health effects of two or more interventions (I), the ROBINS-I tool will be used [39]. It evaluates the risk of bias across confounding, participant selection, classification of the interventions, deviations from intended interventions, missing data, outcome measurements, and reported results. Judgement is rated as "Critical", "Serious", "Moderate", and "Low" risk of bias.

The Critical Appraisal Skills Programme (CASP) will be used for qualitative studies (https://casp-uk.net/casp-tools-checklists). The CASP checklist consists of nine closed questions (e.g., "Was there a clear statement of the aims of the research?" with responses "Yes/Can't tell/No") and one open-ended question (e.g., "How valuable is the research?"). It assesses clarity of research aims, study design, recruitment methods, data collection, relationships between participants and researchers, ethical issues, analyses, description of findings and valuableness of the research.

The methodological quality of mixed methods studies will be assessed using the Mixed Methods Appraisal Tool (MMAT) [40]. The MMAT comprises five criteria for each study type: qualitative, quantitative RCTs, quantitative non-randomized, quantitative descriptive, and mixed methods. All items will be rated as "Yes", "No" or "Can't tell" [41]. Studies will not be excluded based on poor quality, however, any methodological issues will be recorded and highlighted.

## 2.7. Data synthesis

There is no consensus on the best approach for data synthesis in systematic reviews of qualitative and quantitative evidence; this depends on (1) the type/number of included studies and (2) the form/nature of the research question [42]. As recommended by the Cochrane Qualitative and Implementation Methods Group, the final decision about the most appropriate data synthesis method will be made after data extraction and quality assessment [43].

If both quantitative and qualitative studies are included, a results-based convergent synthesis design is planned, with independent syntheses of quantitative and qualitative data, followed by integration of evidence derived from both

syntheses in a third synthesis, in accordance with Joanna Briggs Institute methodology for mixed-methods systematic reviews [42].

Although most included studies are unlikely to be quantitative, if sufficient quantitative data are available, a random-effects meta-analysis will be conducted to synthesise group means and standard deviations from individual studies using the R package meta (https://www.r-project.org/) [44]. Meta-analysis will be performed only if two or more eligible studies report similar or comparable outcomes [45]. Unlike a fixed-effects model, this approach accounts for statistical heterogeneity ($I^2$) among the results of the included studies [46,47]. Funnel plots will be used to explore the possibility of reporting biases and small-study effects where data are available from 10 studies or more [48]. If there are insufficient comparable studies, the synthesis will proceed without a meta-analysis. In this case, findings will be synthesised narratively, with attention to baseline, post-exposure/intervention values, and changes over time, including modality-specific considerations (e.g., circulating vs. neurophysiological biomarkers, eye-tracking markers).

Additionally, where sufficient data are reported and sample size allows, subgroup analyses will be considered to explore potential differences in outcomes according to formal anxiety diagnosis, receipt of therapy for sensory processing challenges, age, and co-occurring conditions among autistic people.

## 2.8. Ethics and dissemination

Ethics approval is not applicable for this study since we will not collect original data. The findings will be disseminated through peer-reviewed publication and conference presentations. Accessible summaries will be shared with autistic individuals, their caregivers, and relevant advocacy groups to ensure the research reaches those who may benefit most.

## 3. Discussion

Current HC-BE does not accommodate the specific needs of autistic children and adults, such as sensory needs, predictability and acceptance [13]. Research comparing hospital guidelines for diverse disabilities and how HC-BEs are perceived by individuals living with disabilities reveals that hospital environments generally fall short in terms of inclusivity. Furthermore, current regulations across various countries focus predominantly on mobility, underscoring the need for updates to better address the broader inclusion requirements [49]. While adopting accessible design has become more common [18], it usually involves designing and building to meet the minimum requirements laid out in regulations, often prioritising accessibility for physically disabled individuals and overlooking those living with "invisible" disabilities, such as ASD. Designing to ensure the building is as inclusive as possible, referred to as inclusive design, has not yet been widely embraced, and research remains limited [19].

HC architecture has addressed the influence of HC space on the health of vulnerable people, using several theoretical models, including Generative Space, [50] salutogenesis and evidence-based design as the most historically influential, aspiring to introduce design elements that alleviate clinical aesthetics [51]. Most of these studies do not differentiate between neurodiverse and neurotypical populations, even though different health conditions or disorders might require targeted environmental adjustments [52]. Normalisation Theory has been instrumental in shifting care from institutional to community-based models, promoting independence and social integration for individuals with mental health challenges or disabilities [53]. It enabled the transition of autistic individuals from specialised institutions to smaller, more integrated community-based, supportive environments. This shift sparked discussions on the optimal settings—whether homes, hostels, small care homes, or schools—and how these domestic, educational and work settings could be adapted to better support them [54].

Regarding specialised HC needs, however, the same level of discourse has not really happened. From the Normalisation Theory prism, autistic individuals would receive care in hospitals similarly to any other patient with the same physiological condition. This would remove the need for clinicians to travel to specialised autism institutions, or for autistic

individuals to visit hospitals via such institutions, thereby reducing the involvement of the institutions overseeing their hospitalisation. Yet, the crucial question of whether hospitals are adequate to receive autistic individuals as patients living independently, or if adaptations are required to support smooth care and recovery of those patients, has not been thoroughly explored.

The finding that autistic individuals encounter more obstacles than the general population in accessing HC-BE emphasises the need to incorporate considerations into the design of such spaces. In addition, given that they may perceive the sensory environment differently from neurotypical individuals, it is crucial to grasp these distinctions to inform inclusive design practices. Although many papers provide recommendations for optimising HC services for autistic individuals, a much smaller body of literature has evaluated interventions aimed at improving access to or experiences of HC-BE for autistic persons [15].

This will be the first systematic review to evaluate and synthesise quantitative and qualitative results on this research topic by using a multi-method approach that integrates a behavioural indices and non-invasive physiological biomarker-based framework. This approach will support a breadth and depth of understanding that confirm or dispute evidence about the HC-BE features and response to anxiety, stress and sensory processing in ASD. The results will also provide valuable information regarding HC-BE intervention components for autistic patients, with implications for researchers, BE practitioners, HC providers and policymakers.

## Supporting information

**S1 Appendix. PRISMA-P (Preferred Reporting Items for Systematic review and Meta-Analysis Protocols) 2015 checklist: Recommended items to address in a systematic review protocol.**
(PDF)

**S2 Appendix. Definition of "healthcare built environment" as used in this review.**
(PDF)

**S3 Appendix. Complete search strategy used in the systematic review for Medline (Pubmed interface).**
(PDF)

**S4 Appendix. Variables collected in pre-established data extraction template.**
(PDF)

## Author contributions

**Conceptualization:** Billie Weaver, Evangelia Chrysikou, Eva Hernandez-Garcia.

**Formal analysis:** Billie Weaver, Eva Hernandez-Garcia.

**Investigation:** Billie Weaver, Mar García-Rodríguez, Daryia Palityka.

**Methodology:** Billie Weaver, Evangelia Chrysikou, Eva Hernandez-Garcia.

**Project administration:** Evangelia Chrysikou, Eva Hernandez-Garcia.

**Supervision:** Evangelia Chrysikou, Eva Hernandez-Garcia.

**Validation:** Evangelia Chrysikou.

**Visualization:** Billie Weaver.

**Writing – original draft:** Billie Weaver.

**Writing – review & editing:** Billie Weaver, Evangelia Chrysikou, Mar García-Rodríguez, Daryia Palityka, Eva Hernandez-Garcia.

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
