## [Decision Letter · Decision Letter 0]

23 Jul 2025

Dear Dr. Hernandez-Garcia,

We look forward to receiving your revised manuscript.

Kind regards,

Jerritta Selvaraj

Academic Editor

PLOS ONE

Additional Editor Comments (if provided):

Reviewers' comments:

Reviewer's Responses to Questions

**Comments to the Author**

1. Does the manuscript provide a valid rationale for the proposed study, with clearly identified and justified research questions?

Reviewer #1: Yes

Reviewer #2: No

2. Is the protocol technically sound and planned in a manner that will lead to a meaningful outcome and allow testing the stated hypotheses?

Reviewer #1: Yes

Reviewer #2: No

3. Is the methodology feasible and described in sufficient detail to allow the work to be replicable?

Reviewer #1: Yes

Reviewer #2: No

4. Have the authors described where all data underlying the findings will be made available when the study is complete?

The PLOS Data policy requires authors to make all data underlying the findings described in their manuscript fully available without restriction, with rare exception, at the time of publication. The data should be provided as part of the manuscript or its supporting information, or deposited to a public repository. For example, in addition to summary statistics, the data points behind means, medians and variance measures should be available. If there are restrictions on publicly sharing data—e.g. participant privacy or use of data from a third party—those must be specified.requires authors to make all data underlying the findings described in their manuscript fully available without restriction, with rare exception, at the time of publication. The data should be provided as part of the manuscript or its supporting information, or deposited to a public repository. For example, in addition to summary statistics, the data points behind means, medians and variance measures should be available. If there are restrictions on publicly sharing data—e.g. participant privacy or use of data from a third party—those must be specified.

Reviewer #1: Yes

Reviewer #2: No

5. Is the manuscript presented in an intelligible fashion and written in standard English?

Reviewer #1: Yes

Reviewer #2: No

You may also provide optional suggestions and comments to authors that they might find helpful in planning their study.

Reviewer #1: The integration of behavioural and physiological markers in the context of healthcare built environments (HC-BE) and autism spectrum disorder (ASD) is an important study which offers potential for novel insights.

Comments

1. The objective as last paragraph in Introduction section will give clarity to the readers

2. It will be interesting to see how the qualitative and quantitative findings would be integrated

3. 10 Biomarkers have been used. The philosophical rationale behind why eye-tracking and salivary cortisol were prioritized over others would strengthen methodological foundation

4. Use of terminologies should be consistent eg. HC - BE

5. Acronyms (e.g., EEG, MEG) appears in tables without being defined in the main text.

6. Grammatical and typographical errors have to be checked for

7. A summary Review Literature Review table bringing out the gaps would add value

Reviewer #2: 1 Synthesize the findings of multiple research studies on this topic, using a rigorous and transparent methodology .Include more studies on this work.

2 Avoid using we, I etc in paragraphs eg:-“ We will search for peer-reviewed articles in PubMed”, Systematic Reviews and MetaAnalyses guidelines will be followed , “Risk of bias will be” etc……

3 You have generalised the Built Environment studies and physiological indicators, elaborate the work on each environment given in literature and cite the reference.

4 I find more redundancy in the manuscript.

5 You have just specified the name of biomarkers used in the study , Need to explain how it is used , outcome etc and should have complete information about each biomarker.

6 Table 3 is repeated content from the paragraph. What the reader can infer from the table??

7 Which author extract the data? Are you reviewing a work done in literature study “Data extraction will be conducted by one author (BW)”-Page 11.

8 Study Selection:- Need to know how to synthesize the study not how you use reference manager.

9 Expand the abbreviation at the beginning of the passage followed by its short form in rest of the paragraph.

.

Reviewer #1: **Yes:** Arun SahayadhasArun SahayadhasArun SahayadhasArun Sahayadhas

Reviewer #2: No

---

## [Author Response · Author response to Decision Letter 1]

6 Sep 2025

RESPONSE TO REVIEWER #1 (PONE-D-25-32161)

Point 1: The objective as last paragraph in Introduction section will give clarity to the readers.

Response 1: We thank the reviewer for this comment. We have added a more concise objective paragraph at the end of the Introduction to provide greater clarity regarding the aims of the review. The revised text now reads: “This systematic review aims to explore whether the exposure to specific HC-BE features can alter behavioural and physiological indicators of anxiety, stress, and emotion in ASD, which is associated with atypical sensory processing, integrating both qualitative and quantitative evidence to identify effective design strategies.”

Point 2: It will be interesting to see how the qualitative and quantitative findings would be integrated.

Response 2: We thank the reviewer for this insightful comment. We agree that integrating qualitative and quantitative findings is crucial in our systematic review. This approach was already described in the manuscript, Methods/Data Synthesis section: “If there is a combination of quantitative and qualitative studies, a results-based convergent synthesis design is planned, with independent syntheses of quantitative and qualitative data, followed by integration of evidence derived from both syntheses in a third synthesis in accordance with Joanna Briggs Institute methodology for conducting a mixed-methods systematic review [40].” We also believe that our planned approach will provide a clear and rigorous framework for combining multiple types of evidence.

Point 3: 10 Biomarkers have been used. The philosophical rationale behind why eye-tracking and salivary cortisol were prioritized over others would strengthen methodological foundation.

Response 3: We thank the reviewer for this comment. We would like to clarify that in our manuscript all ten non-invasive biomarkers listed in Table 2 are considered equally; no emphasis or prioritization is placed on eye-tracking or salivary cortisol over the other biomarkers. In the manuscript, we clearly stated that these ten biomarkers were selected because they showed the most consistent results across systematic reviews, which follow rigorous methodology. This selection approach ensures that all biomarkers are included on an equal basis. We have additionally clarified this point in the Outcomes section to avoid any potential misunderstanding regarding the selection of biomarkers with the following sentence: “All ten non-invasive biomarkers were included based on their consistent results in prior systematic reviews, ensuring a rigorous and unbiased selection.”

Point 4: Use of terminologies should be consistent eg. HC – BE.

Response 4: We thank the reviewer for this comment. We have carefully reviewed the manuscript and ensured that the terminology, particularly “HC-BE”, is now used consistently throughout the text, tables, and figures.

Point 5: Acronyms (e.g., EEG, MEG) appears in tables without being defined in the main text.

Response 5: We thank the reviewer for this comment. We have double-checked the main text and tables to ensure that each acronym matches its first definition and that the abbreviations are used consistently throughout the manuscript.

Point 6: Grammatical and typographical errors have to be checked for.

Response 6: We thank the reviewer for pointing this out. We have carefully reviewed the manuscript for grammatical and typographical errors and corrected them to ensure clarity and readability throughout the text and tables.

Point 7: A summary Review Literature Review table bringing out the gaps would add value.

Response 7: We thank the reviewer for this thoughtful suggestion. We fully agree that a summary table of the existing literature can often add value in a manuscript. In this case, however, we are presenting a protocol for a mixed-methods systematic review. The primary aim of the protocol is to outline the methodology that will be applied to systematically identify, appraise, and synthesize the existing evidence. Preparing a literature review table at this stage, before conducting the systematic search, screening, and appraisal steps, could introduce bias regarding which studies, gaps, populations, or outcomes are highlighted. Since the systematic review itself will provide a rigorous and transparent synthesis of the literature, including such a table at the protocol stage would risk being redundant and potentially misleading. We have carefully considered the reviewer’s suggestion but have chosen not to include a review summary table in the current manuscript to preserve methodological rigor and avoid providing readers with potentially biased information. Instead, the forthcoming mixed-methods systematic review will incorporate a structured data extraction process, as outlined in the Data Extraction section, and will present tabular summaries of findings where appropriate; the most suitable synthesis approach will be determined after data extraction and quality assessment, as described in the Data Synthesis section of the manuscript.

RESPONSE TO REVIEWER #2 (PONE-D-25-32161)

Point 1: Synthesize the findings of multiple research studies on this topic, using a rigorous and transparent methodology . Include more studies on this work.

Response 1: We thank the reviewer for this comment and fully agree that synthesizing findings from multiple studies using a rigorous and transparent methodology is essential. We would like to clarify, however, that the present manuscript is a protocol for a mixed-methods systematic review. Its purpose is to describe in detail the methodology that will be applied to systematically identify, appraise, and synthesise the relevant studies following the screening process. The actual synthesis of findings and inclusion of all eligible studies will be conducted in the subsequent systematic review, in accordance with the procedures outlined in this protocol.

Point 2: Avoid using we, I etc in paragraphs eg:-“ We will search for peer-reviewed articles in PubMed”, Systematic Reviews and MetaAnalyses guidelines will be followed , “Risk of bias will be” etc……

Response 2: We thank the reviewer for this helpful comment. We recognize that excessive use of first-person pronouns (e.g., “we will…”) can detract from a neutral and objective tone in scientific writing. While PLOS ONE permits the use of first-person pronouns, we agree that greater neutrality is preferable when describing the methodology of the protocol. To address this, we have revised the manuscript to limit first-person usage specifically in sections outlining methodological procedures (e.g., changed “We will search for peer-reviewed articles in PubMed” to “Peer-reviewed articles will be searched in PubMed”). This adjustment ensures that the methodological descriptions remain objective and rigorous, while retaining clarity and readability in other sections of the manuscript.

Point 3: You have generalised the Built Environment studies and physiological indicators, elaborate the work on each environment given in literature and cite the reference.

Response 3: We appreciate the reviewer’s comment. In the Introduction, we have added specific details on which BE features were studied in prior systematic reviews and, where available, their links to corresponding physiological and behavioural biomarkers. We would like to clarify, however, that the primary aim of this protocol is to systematically link HC-BE-specific details with the most consistent biomarkers identified in previous studies. To select the appropriate biomarkers, we consulted systematic reviews of clinical data. The protocol outlines how the review will extract and synthesise HC-BE-specific information associated with physiological and behavioural biomarkers. For each type of HC-BE (e.g., hospital wards, outpatient clinics, sensory rooms), the review will systematically record details on HC-BE features and their evaluation alongside the corresponding biomarkers. This process is fully described in the Data Extraction and Data Synthesis sections of the manuscript.

Point 4: I find more redundancy in the manuscript.

Response 4: We acknowledge the reviewer’s observation regarding perceived redundancy. To address this, we carefully reviewed the entire manuscript and implemented several revisions to enhance conciseness and readability:

1. Introduction and Discussion: Merged sentences that previously described HC-BE features, physiological and behavioural biomarkers, and their interrelations, using more concise and streamlined phrasing.

2. Methodology: Replaced repeated restatements of the study aim and methodological details with references to the appropriate sections, thereby reducing repetition while maintaining clarity.

3. Tables: Reviewed all tables and corresponding main text passages, removing duplicated explanations to avoid redundancy and improve readability.

These changes ensure the manuscript presents the scientific content in a coherent, logically structured manner without unnecessary repetition.

Point 5: You have just specified the name of biomarkers used in the study , Need to explain how it is used , outcome etc and should have complete information about each biomarker.

Response 5: We appreciate the reviewer’s comment. In the revised manuscript, we have expanded the Outcomes section to provide detailed information about each biomarker. Specifically, we now describe how each biomarker is assessed and how outcomes will be measured (e.g., baseline levels, pre–post intervention changes, or cut-off thresholds where applicable). This detailed information informs the planning of data synthesis for quantitative outcomes, and we have added a corresponding description in the Data Synthesis section. These revisions ensure that the Outcomes and Data Synthesis sections are fully aligned with PRISMA guidelines and enhance transparency for future replication.

Point 6: Table 3 is repeated content from the paragraph. What the reader can infer from the table?

Response 6: We thank the reviewer for highlighting this point. After careful consideration, we agree with the reviewer that Table 3 does not present additional, unique information beyond the main text. We have now provided a detailed and coherent description of the biomarkers and behavioural measures, and have applied the same approach for Table 2. Therefore, to avoid redundancy and maintain readability, we have decided not to include Tables 2 and 3.

Point 7: Which author extract the data? Are you reviewing a work done in literature study “Data extraction will be conducted by one author (BW)”-Page 11.

Response 7: In this systematic review, data extraction will be conducted by one author (BW) and then verified using a crowdsourcing approach. We have now explicitly specified the members of the review team who will verify the data: Daryia Palitika (DP) and Mar Garcia Rodriguez (MGR). The crowdsourcing approach is widely accepted in systematic reviews when clearly reported, as it ensures accuracy and consistency. This method also allows the inclusion of additional members, identified via Cochrane Task Exchange, to assist with non-English articles, supporting a rigorous and reliable synthesis of the literature.

Point 8: Study Selection:- Need to know how to synthesize the study not how you use reference manager.

Response 8: We thank the reviewer for the comment. The procedures for synthesizing the included studies are already detailed in the manuscript. In the Data Extraction section, we describe the variables to be collected and how data will be verified. In the Data Synthesis section, we explain the planned integration of quantitative and qualitative findings using a results-based convergent synthesis design, and the potential use of random-effects meta-analysis for quantitative outcomes. The information provided in the Study Selection section follows PRISMA guidelines, which require a clear description of the process used to identify and screen studies. The mention of the reference manager refers only to organizational purposes and is not part of the synthesis process.

Point 9: Expand the abbreviation at the beginning of the passage followed by its short form in rest of the paragraph.

Response 9: We acknowledge the reviewer’s observation. We have now revised the manuscript to ensure that all abbreviations are expanded at first mention and consistently abbreviated thereafter.

---

## [Decision Letter · Decision Letter 1]

9 Dec 2025

Dear Dr. Hernandez-Garcia,

Thank you for submitting your manuscript to PLOS One. I have received feedback from three experts in the field. They all agree that your manuscript has merit, but that it will require revision to be suitable for publication. Each provides excellent suggestions which I will not restate here. However, I encourage you to respond to each reviewer’s comments carefully. Also, please carefully consider where to place information (e.g., introduction vs discussion).

We look forward to receiving your revised manuscript.

Kind regards,

Eric J. Moody, Ph.D.

Academic Editor

PLOS One

Journal Requirements:

Reviewers' comments:

Reviewer's Responses to Questions

**Comments to the Author**

1. Does the manuscript provide a valid rationale for the proposed study, with clearly identified and justified research questions?

Reviewer #3: Yes

Reviewer #4: Partly

Reviewer #5: Yes

2. Is the protocol technically sound and planned in a manner that will lead to a meaningful outcome and allow testing the stated hypotheses?

Reviewer #3: Yes

Reviewer #4: Partly

Reviewer #5: Yes

3. Is the methodology feasible and described in sufficient detail to allow the work to be replicable?

Reviewer #3: Yes

Reviewer #4: Yes

Reviewer #5: Yes

4. Have the authors described where all data underlying the findings will be made available when the study is complete?

The PLOS Data policy requires authors to make all data underlying the findings described in their manuscript fully available without restriction, with rare exception, at the time of publication. The data should be provided as part of the manuscript or its supporting information, or deposited to a public repository. For example, in addition to summary statistics, the data points behind means, medians and variance measures should be available. If there are restrictions on publicly sharing data—e.g. participant privacy or use of data from a third party—those must be specified.requires authors to make all data underlying the findings described in their manuscript fully available without restriction, with rare exception, at the time of publication. The data should be provided as part of the manuscript or its supporting information, or deposited to a public repository. For example, in addition to summary statistics, the data points behind means, medians and variance measures should be available. If there are restrictions on publicly sharing data—e.g. participant privacy or use of data from a third party—those must be specified.

Reviewer #3: No

Reviewer #4: No

Reviewer #5: No

5. Is the manuscript presented in an intelligible fashion and written in standard English?

Reviewer #3: No

Reviewer #4: Yes

Reviewer #5: Yes

You may also provide optional suggestions and comments to authors that they might find helpful in planning their study.

Reviewer #3: Major part of the discussion is focussed on built environment design and briefly on the impact of the behaviour of the children of autism. Further the language used in discussion is in future tense .No conclusive results shown.

Limitations of the study not mentioned. No future directions given.

No dedicated section on Results. There is no mention of the documentation of the research process in log document.

There is no mention of how many research papers were reviewed and of which how many were potentially relevant articles were retrieved. There is no summary of person characteristics of behaviour to understand the impact of built environment.

Reviewer #4: The authors present a protocol for a systematic review pertaining to how different factors modulate the well-being, emotional or stress response (assessed at the behavioral and physiological levels) in healthcare environments, for autistic persons of all age and profile. I think the final work could be really valuable. The manuscript is overall well written. But I do have major and minor concerns, listed below.

Major concerns

1. Page 4: “Interestingly, a state of hyperarousal evaluated through salivary cortisol (CT) in autism demonstrates correlated associations with heightened indices of behavioural anxiety/stress response [21].”

I think that this sentence is confusing and/or misleading. In the article cited by the authors, “heightened repetitive language (subscale of RBQ) was positively correlated with the Total Anx-DOS Score (rs (22) = 0.452, p = .027)”, a behavioral index of anxiety. “There were no significant correlations between Total Anx-DOS score and subscale scores from the SRS-2 or RBQ.” Meaning that the association mentioned above was only valid for one of the two neurogenetic disorders included in the study. No association was found in the study between Anx-DOS and salivary cortisol. So in all I think this sentence should be removed or modified to better reflect what was found or not in the original study.

Crawford, H., Oliver, C., Groves, L., Bradley, L., Smith, K., Hogan, A., Renshaw, D., Waite, J., & Roberts, J. (2023). Behavioural and physiological indicators of anxiety reflect shared and distinct profiles across individuals with neurogenetic syndromes. Psychiatry Research, 326, 115278. https://doi.org/10.1016/j.psychres.2023.115278

2. Page 7: “Both inpatient and outpatient HC-BE will be included, however, those purposively designed for autistic users will not be considered”

I don’t understand this choice. Why not treat these studies as a subgroup of data to compare with environments that have not been purposively designed for autistic users? I think this could lead to very interesting results on what specific aspects of BE affect autistic persons’ preferences and well-being? Please provide a rationale for this choice.

3. Page 8: “EEG measures cortical electrical activity across frequency bands (e.g., alpha, beta, theta), while MEG detects rapid changes in brain responses to sensory stimuli.”

I don’t think that this sentence reflects accurately what EEG and MEG can tell us about brain activity. The main difference between the two methods is in the nature of the signal: electrical activity measured through captors placed on the scalp for EEG, reflecting the neurons’ action potentials; subtle changes in the magnetic field for MEG, also reflecting the neuron’s action potentials. I don’t think that these two measures differ in how they capture spontaneous or evoked activity. They both can do that. Please correct this sentence accordingly.

4. Page 4: ”impact neural pathways and structures of emotional processing [19].”

The notion of neural pathways in my opinion reflect physiological data pertaining to connections in the brain, such as Diffusion Tensor Imaging. Is this what is intended by the authors? Maybe simply replace “pathways” by “activity” to reflect that you aggregated data pertaining to structural and functional imaging?

5. Discussion section.

Most of the paragraphs from this section seem like they could also be placed in the Introduction section, except for the final paragraph. I don’t know if it’s a problem or not. I don’t really know what is expected in the Discussion section of an article presenting a review research protocol, since no results exist yet... Please refer to the Editor decision regarding this concern.

Minor concerns

1. Abstract: “A growing body of systematic studies links autism-friendly BEs with positive care experiences, yet substantial gaps remain in understanding the effects on behavioural and physiological responses of emotion”

I think the formulation “the effects on behavioural and physiological aspects of emotional responses” would be more exact. “Responses of emotion” is not really a valid statement in my opinion.

2. Page 3: “The HC built environment (HC-BE) affects vulnerable individuals and influences patients' perceptions and sense of agency in ways that may not impact those who self-identify as healthy [7] and this might not be an exception for autistic people as explored in this review”

I find this sentence very complex to understand. Maybe use this formulation instead?: “While the HC built environment (HC-BE) may not impact much those who self-identify as healthy [7], it could affect vulnerable individuals and influence patients' perceptions and sense of agency, and this might not be an exception for autistic people as explored in this review”.

3. Page 3: “This effect may occur across intimate, private, and public HC-BE settings”

I understand what private and public HC-BE settings could be. But what could be an intimate HC-BE setting?

4. Page 5: Please explain the PI/ECOS acronym at first use.

Page 8: Same for Salivary CT

5. Page 8: typo. There is a missing bullet point in front of the “behavioral” section.

6. Page 11: “(c) participant, patient characteristics: sample size, participant characteristics, eligibility criteria ».

« Participant characteristics » is repeated twice.

7. Page 14 : « In this case, findings will be synthesised narratively, with with attention to baseline”.

Please remove one “with”.

8. Page 15: “This would negate the need for clinicians to visit the person requiring care at a specialised autism institution or for autistic individuals to visit the hospitals from the institutions, and therefore the institution overseeing their hospitalization”

I don’t understand this sentence. Could you please rephrase it?

Reviewer #5: This a well designed protocol, and the authors have been responsive to the first round of Reviewers' comments. There are a few things that the authors need to address before the manuscript can be accepted for publication:

1. Data availability statement should be part of the manuscript.

2. The need for this review stems from the fact that autistic individuals experience anxiety (which includes stress) as well as sensory processing. What this protocol has failed to account for is that there are degrees of anxiety as well as degrees of sensory processing. The results of this review should evaluate the results for autistic individuals diagnosed with an anxiety disorder vs. autistic individuals not diagnosed with an anxiety disorder (that most certainly experience a certain level of anxiety in HC settings), as well as the results for autistic individuals that have received therapy for a sensory processing disorder (since diagnosis is not recognized in DSM-5) vs. autistic individuals that have not received therapy for a sensory processing disorder (that might experience a certain level of sensory issues in HC settings). This info may not be available in all studies, but it is bound to be present in quite a few. Other planned subgroup analyses (e.g., by age, co-occurring conditions, or cognitive level) should also be included in the protocol to address heterogeneity in the ASD population.

3. Somewhat related to point 2 above, since the underpinning of this review is anxiety and sensory processing, it is important to mention in the introduction what % of autistic individuals are diagnosed with an anxiety disorder as well as what % typically undergo therapy for sensory processing. This should be framed in the context that the rest of the autistic individuals do experience anxiety, which may not typically reach clinical levels (hence the lack of diagnosis) - and a similar point for sensory processing.

4. It is admirable that the authors chose to use identity first in this protocol. For consistency, in Table 1, please rephrase "patients with ASD" which is the same as "patients with autism" with "autistic patients" or "patients diagnosed with ASD." Similarly, in section 3. Discussion, please replace "those with physical disabilities" with physically disabled individuals."

5. People that are meant to ultimately benefit from the results of these study (autistic individuals, caregivers of autistic individuals) should be included in some manner in the dissemination of the results - to be included in Section 2.8.

.

Reviewer #3: **Yes:** Shabina AhmedShabina AhmedShabina AhmedShabina Ahmed

Reviewer #4: **Yes:** Matias BaltazarMatias BaltazarMatias BaltazarMatias Baltazar

Reviewer #5: No

---

## [Author Response · Author response to Decision Letter 2]

23 Jan 2026

RESPONSE TO REVIEWER #3

Comment 1: Major part of the discussion is focussed on built environment design and briefly on the impact of the behaviour of the children of autism. Further the language used in discussion is in future tense .No conclusive results shown.

Limitations of the study not mentioned. No future directions given.

No dedicated section on Results. There is no mention of the documentation of the research process in log document.

There is no mention of how many research papers were reviewed and of which how many were potentially relevant articles were retrieved. There is no summary of person characteristics of behaviour to understand the impact of built environment.

Response 1: We would like to clarify that the submitted manuscript is a protocol for a mixed‑methods systematic review, not a completed review reporting findings. The manuscript has been developed following PRISMA‑Protocols guidelines. As a protocol, several elements noted by the reviewer, such as the number of included studies, participant characteristics, and results, are not expected at this stage. The use of future tense throughout reflects the planned methodology.

The review aims to examine how features of the HC-BE influence behavioural and physiological indicators of stress responses in autistic individuals. Accordingly, the Discussion presents relevant aspects of the HC-BE on this topic, their theoretical and historical context, and the anticipated implications for this field, while acknowledging that the actual impact on outcomes will only be determined once the review is conducted.

RESPONSE TO REVIEWER #4

Major concerns

Point 1: Page 4: “Interestingly, a state of hyperarousal evaluated through salivary cortisol (CT) in autism demonstrates correlated associations with heightened indices of behavioural anxiety/stress response [21].”

I think that this sentence is confusing and/or misleading. In the article cited by the authors, “heightened repetitive language (subscale of RBQ) was positively correlated with the Total Anx-DOS Score (rs (22) = 0.452, p = .027)”, a behavioral index of anxiety. “There were no significant correlations between Total Anx-DOS score and subscale scores from the SRS-2 or RBQ.” Meaning that the association mentioned above was only valid for one of the two neurogenetic disorders included in the study. No association was found in the study between Anx-DOS and salivary cortisol. So in all I think this sentence should be removed or modified to better reflect what was found or not in the original study.

Crawford, H., Oliver, C., Groves, L., Bradley, L., Smith, K., Hogan, A., Renshaw, D., Waite, J., & Roberts, J. (2023). Behavioural and physiological indicators of anxiety reflect shared and distinct profiles across individuals with neurogenetic syndromes. Psychiatry Research, 326, 115278. https://doi.org/10.1016/j.psychres.2023.115278

Response 1: We thank the reviewer for this important clarification and agree that the original wording was potentially misleading. In response, we have revised the sentence to remove references to hyperarousal and direct behavioural–physiological coupling, and to more accurately reflect the syndrome-specific and heterogeneous nature of the findings. The revised text now emphasizes altered basal physiological arousal in neurogenetic syndromes linked to autistic phenotypes without overstating associations with behavioural anxiety measures.

Point 2: Page 7: “Both inpatient and outpatient HC-BE will be included, however, those purposively designed for autistic users will not be considered”

I don’t understand this choice. Why not treat these studies as a subgroup of data to compare with environments that have not been purposively designed for autistic users? I think this could lead to very interesting results on what specific aspects of BE affect autistic persons’ preferences and well-being? Please provide a rationale for this choice.

Response 2: Our decision to exclude care environments purposively designed for autistic users is grounded in the conceptual aims of our review. Since most autistic individuals receive care in standard HC facilities rather than specialist units, our focus is on understanding their real‑world experiences within care environments that were not designed with autism in mind.

It is important to clarify that, to our best knowledge, national or international HC design guidance—including comprehensive frameworks such as the UK Health Building Notes—does not include autism‑specific accessibility requirements. This gap is also reflected in HC design guidance reviewed in other research projects across multiple countries. Consequently, any HC facility not explicitly created as an autism‑specific unit cannot be considered “purpose‑built” for autistic users. Mainstream HC settings such as general hospitals, maternity units, dental clinics, primary care facilities, and most psychiatric wards are designed for the general population, with the needs of autistic people absent from the design briefs that inform these environments.

Including specialist autism‑designed settings would therefore introduce a fundamentally different design context and fall outside our aim of identifying BE features that affect autistic individuals in the standard HC facilities they encounter across their lifespan.

Point 3: Page 8: “EEG measures cortical electrical activity across frequency bands (e.g., alpha, beta, theta), while MEG detects rapid changes in brain responses to sensory stimuli.”

I don’t think that this sentence reflects accurately what EEG and MEG can tell us about brain activity. The main difference between the two methods is in the nature of the signal: electrical activity measured through captors placed on the scalp for EEG, reflecting the neurons’ action potentials; subtle changes in the magnetic field for MEG, also reflecting the neuron’s action potentials. I don’t think that these two measures differ in how they capture spontaneous or evoked activity. They both can do that. Please correct this sentence accordingly.

Response 3: We appreciate the reviewer’s careful reading and agree that our original sentence did not accurately characterise the distinction between EEG and MEG. We have revised the sentence in the manuscript to more accurately reflect the nature of these neurophysiological measures, as follows:

“EEG and MEG both measure neural activity arising from synchronous postsynaptic currents, with EEG detecting electrical potentials at the scalp and MEG detecting the associated magnetic fields; both methods can record spontaneous and evoked activity.”

Point 4: Page 4: ”impact neural pathways and structures of emotional processing [19].”

The notion of neural pathways in my opinion reflect physiological data pertaining to connections in the brain, such as Diffusion Tensor Imaging. Is this what is intended by the authors? Maybe simply replace “pathways” by “activity” to reflect that you aggregated data pertaining to structural and functional imaging?

Response 4: We thank the reviewer for this suggestion. To avoid potential misinterpretation that the term “neural pathways” refers only to anatomical connections, we have replaced it with “neural activity”. The revised sentence now reads:

“impact neural activity and structures of emotional processing [19].”

Point 5: Discussion section.

Most of the paragraphs from this section seem like they could also be placed in the Introduction section, except for the final paragraph. I don’t know if it’s a problem or not. I don’t really know what is expected in the Discussion section of an article presenting a review research protocol, since no results exist yet... Please refer to the Editor decision regarding this concern.

Response 5:

We thank the reviewer for this comment. We note that in a systematic review protocol, the Discussion section often overlaps with the Introduction, as there are no results yet to interpret. According to PRISMA-P (Item 15) guidance, the Discussion in a protocol (an optional and forward-looking section) is primarily intended to anticipated contributions and significance, and potential implications of the planned review. Our Discussion has been structured accordingly to: (i) emphasise the current evidence to highlight the expected contributions and the need for multi-method assessment of behavioural and physiological responses, (ii) provide historical and theoretical context relevant to the field, essential for contextualising the review, and (iii) outline the expected value and implications of the planned review (for research, practice, and policy). We think this structure is consistent with PRISMA-P recommendations for review protocols.

According to PRISMA-P (Items 6–8), the Introduction provides the key background and rationale for the review, identifies the main gaps in existing knowledge, clearly defines the systematic review question (framed in PI/ECO(S) terms), and provides brief context for eligibility criteria.

Reference: Moher D, et al. PRISMA-P 2015 statement. Systematic Reviews. 2015;4:1.

Minor concerns

Point 1: Abstract: “A growing body of systematic studies links autism-friendly BEs with positive care experiences, yet substantial gaps remain in understanding the effects on behavioural and physiological responses of emotion”

I think the formulation “the effects on behavioural and physiological aspects of emotional responses” would be more exact. “Responses of emotion” is not really a valid statement in my opinion.

Response 1: we thank the reviewer for this comment. We have revised the wording to improve conceptual clarity, replacing “responses of emotion” with “behavioural and physiological aspects of emotional responses.”

Point 2: Page 3: “The HC built environment (HC-BE) affects vulnerable individuals and influences patients' perceptions and sense of agency in ways that may not impact those who self-identify as healthy [7] and this might not be an exception for autistic people as explored in this review”

I find this sentence very complex to understand. Maybe use this formulation instead?: “While the HC built environment (HC-BE) may not impact much those who self-identify as healthy [7], it could affect vulnerable individuals and influence patients' perceptions and sense of agency, and this might not be an exception for autistic people as explored in this review”.

Response 2: We thank the reviewer for this helpful suggestion. Upon review, we agree that the original sentence may have been overly complex. We have revised the wording to improve readability while keeping the original meaning. The revised sentence now reads:

“While the HC built environment (HC-BE) may have limited impact on those who self-identify as healthy [7], it can affect vulnerable individuals and influence patients’ perceptions and sense of agency, and this might not be an exception for autistic people as explored in this review.”

Point 3: Page 3: “This effect may occur across intimate, private, and public HC-BE settings”

I understand what private and public HC-BE settings could be. But what could be an intimate HC-BE setting?

Response 3: We thank the reviewer for raising this point. In the HC-BE literature and field, “intimate” settings refer to small-scale, high-proximity clinical spaces where patients experience close interpersonal interaction and heightened vulnerability. This term reflects established usage within the field and captures a dimension of HC space that is particularly relevant when considering stress responses in autistic individuals.

Point 4: Page 5: Please explain the PI/ECOS acronym at first use.

Page 8: Same for Salivary CT

Response 4: The PI/ECOS acronym has now been introduced and defined at first use in the manuscript. Regarding salivary CT, the acronym for salivary cortisol was already defined on page 4.

Point 5: Page 8: typo. There is a missing bullet point in front of the “behavioral” section.

Response 5: We have carefully checked the manuscript and could not identify a missing bullet point before the “Behavioural” section. We have ensured that all bullet points are consistent and correctly formatted throughout the text.

Point 6: Page 11: “(c) participant, patient characteristics: sample size, participant characteristics, eligibility criteria ».

« Participant characteristics » is repeated twice.

Response 6: We thank the reviewer for noticing this repetition. We have revised the text to remove the duplicated phrase.

Point 7: Page 14 : « In this case, findings will be synthesised narratively, with with attention to baseline”.

Please remove one “with”.

Response 7: We have removed the duplicated “with” in the manuscript.

Point 8: Page 15: “This would negate the need for clinicians to visit the person requiring care at a specialised autism institution or for autistic individuals to visit the hospitals from the institutions, and therefore the institution overseeing their hospitalization”

I don’t understand this sentence. Could you please rephrase it?

Response 8: We thank the reviewer for highlighting this sentence. We have revised it for clarity, replacing “negate” with “remove” and simplifying the structure. The revised sentence now reads:

“This would remove the need for clinicians to travel to specialised autism institutions, or for autistic individuals to visit hospitals via such institutions, thereby reducing the involvement of the institutions overseeing their hospitalisation.”

RESPONSE TO REVIEWER #5

Point 1: Data availability statement should be part of the manuscript.

Response 1: We thank the reviewer for this comment. We have introduced a new Data Availability Statement section, which can be found on page 16 of the revised manuscript.

Point 2: The need for this review stems from the fact that autistic individuals experience anxiety (which includes stress) as well as sensory processing. What this protocol has failed to account for is that there are degrees of anxiety as well as degrees of sensory processing. The results of this review should evaluate the results for autistic individuals diagnosed with an anxiety disorder vs. autistic individuals not diagnosed with an anxiety disorder (that most certainly experience a certain level of anxiety in HC settings), as well as the results for autistic individuals that have received therapy for a sensory processing disorder (since diagnosis is not recognized in DSM-5) vs. autistic individuals that have not received therapy for a sensory processing disorder (that might experience a certain level of sensory issues in HC settings). This info may not be available in all studies, but it is bound to be present in quite a few. Other planned subgroup analyses (e.g., by age, co-occurring conditions, or cognitive level) should also be included in the protocol to address heterogeneity in the ASD population.

Response 2: We thank the reviewer for this helpful comment. We would like to clarify that anxiety disorders were already considered within the category of comorbidities in our data extraction plan (Section 2.5, subsection d; page 12). To improve the focus on it, we have now explicitly specified anxiety disorders under the “comorbidities” variable (“clinical characteristics” subsection) in the manuscript. We agree that sensory processing differences are a core feature in our study. Consequently, we have clarified under the same “clinical characteristics” variable that sensory processing characteristics and related interventions (e.g., receipt of therapy targeting sensory processing challenges), where reported, will also be extracted. These clarifications were made to ensure that heterogeneity in anxiety and sensory processing is captured more explicitly.

Furthermore, we have specified that planned subgroup analyses will include anxiety disorder diagnosis versus no formal diagnosis, receipt of therapy for sensory processing challenges versus none, as well as age and co-occurring conditions, to account for heterogeneity within the autistic individuals. To reflect this, a sentence describing the planned subgroup analyses has been added to the Data Synthesis section (Section 2.7; page 14).

The Supporting Information (S4 Appendix) has been updated accordingly.

Point 3: Somewhat related to point 2 above, since the underpinning of this r

---

## [Decision Letter · Decision Letter 2]

17 Mar 2026

Dear Dr. Hernandez-Garcia,

We look forward to receiving your revised manuscript.

Kind regards,

Eric J. Moody, Ph.D.

Academic Editor

PLOS One

**Journal Requirements:**

Reviewers' comments:

Reviewer's Responses to Questions

**Comments to the Author**

1. Does the manuscript provide a valid rationale for the proposed study, with clearly identified and justified research questions?

Reviewer #3: Yes

Reviewer #4: Yes

Reviewer #5: Yes

2. Is the protocol technically sound and planned in a manner that will lead to a meaningful outcome and allow testing the stated hypotheses?

Reviewer #3: Yes

Reviewer #4: Yes

Reviewer #5: Yes

3. Is the methodology feasible and described in sufficient detail to allow the work to be replicable?

Reviewer #3: Yes

Reviewer #4: Yes

Reviewer #5: Yes

4. Have the authors described where all data underlying the findings will be made available when the study is complete?

The PLOS Data policy requires authors to make all data underlying the findings described in their manuscript fully available without restriction, with rare exception, at the time of publication. The data should be provided as part of the manuscript or its supporting information, or deposited to a public repository. For example, in addition to summary statistics, the data points behind means, medians and variance measures should be available. If there are restrictions on publicly sharing data—e.g. participant privacy or use of data from a third party—those must be specified.requires authors to make all data underlying the findings described in their manuscript fully available without restriction, with rare exception, at the time of publication. The data should be provided as part of the manuscript or its supporting information, or deposited to a public repository. For example, in addition to summary statistics, the data points behind means, medians and variance measures should be available. If there are restrictions on publicly sharing data—e.g. participant privacy or use of data from a third party—those must be specified.

Reviewer #3: Yes

Reviewer #4: Yes

Reviewer #5: Yes

5. Is the manuscript presented in an intelligible fashion and written in standard English?

Reviewer #3: Yes

Reviewer #4: Yes

Reviewer #5: Yes

You may also provide optional suggestions and comments to authors that they might find helpful in planning their study.

Reviewer #3: Corrections in the manuscript as required by the reviewers have been made.

However I would suggest a small correction in page 9 the acronyms used for autonomic measures HRV and EDA to be written in expanded form.

There is no information on the limitations of this study.

Reviewer #4: I think that the authors did a great job in answering my concerns. I just have two minor concerns left, listed below, more as ideas to improve the manuscript.

1. I think that the authors’ response to my initial major concern n°2 is really great. Maybe the authors should consider including their response to my comment in the main text of the manuscript? Right after the sentence “Both inpatient and outpatient HC-BE will be included, however, those purposively designed for autistic users will not be considered”, page 7? I copy pasted their answer below.

"Our decision to exclude care environments purposively designed for autistic users is grounded in the conceptual aims of our review. Since most autistic individuals receive care in standard HC facilities rather than specialist units, our focus is on understanding their real world experiences within care environments that were not designed with autism in mind. It is important to clarify that, to our best knowledge, national or international HC design guidance—including comprehensive frameworks such as the UK Health Building Notes—does not include autism specific accessibility requirements. This gap is also reflected in HC design guidance reviewed in other research projects across multiple countries. Consequently, any HC facility not explicitly created as an autism specific unit cannot be considered “purpose built” for autistic users. Mainstream HC settings such as general hospitals, maternity units, dental clinics, primary care facilities, and most psychiatric wards are designed for the general population, with the needs of autistic people absent from the design briefs that inform these environments. Including specialist autism designed settings would therefore introduce a fundamentally different design context and fall outside our aim of identifying BE features that affect autistic individuals in the standard HC facilities they encounter across their lifespan."

2. Regarding my initial minor concern n°3, please provide a short definition or some examples of “intimate, private, and public HC-BE settings” in the main text of the manuscript, page 3, in the Introduction section.

Reviewer #5: The authors have addressed all five points raised in my previous review in a thorough and satisfactory manner. Specifically: (1) a Data Availability Statement has been added to the manuscript body; (2) subgroup analyses stratifying by anxiety disorder diagnosis, receipt of sensory processing therapy, age, and co-occurring conditions have been explicitly incorporated into the manuscript; (3) the Introduction now includes the prevalence of diagnosed anxiety disorders (~35%) and acknowledges that many autistic individuals experience sub-threshold anxiety symptoms as well as sensory processing differences, supported by two new references; (4) identity-first language has been applied consistently, including in Table 1 and the Discussion; and (5) Section 2.8 now explicitly names autistic individuals and their caregivers as stakeholders in dissemination.

I have no further concerns. The protocol is well-structured, methodologically sound, and appropriately follows PRISMA-P guidelines. I recommend acceptance.

.

Reviewer #3: **Yes:** SHABINA AHMEDSHABINA AHMEDSHABINA AHMEDSHABINA AHMED

Reviewer #4: **Yes:** Matias BaltazarMatias BaltazarMatias BaltazarMatias Baltazar

Reviewer #5: No

---

## [Author Response · Author response to Decision Letter 3]

30 Mar 2026

RESPONSE TO REVIEWER #3

Comment 1: Reviewer #3: Corrections in the manuscript as required by the reviewers have been made. However I would suggest a small correction in page 9 the acronyms used for autonomic measures HRV and EDA to be written in expanded form.

There is no information on the limitations of this study.

Response 1: We truly appreciate your careful reading and constructive feedback in our protocol manuscript. We would like to kindly note that the acronyms for the two terms Heart rate variability (HRV) and electrodermal activity markers (EDA) were already introduced on page 8. For this reason, we continued using the acronyms on page 9 to maintain consistency and avoid repetition.

RESPONSE TO REVIEWER #4

Reviewer #4: I think that the authors did a great job in answering my concerns. I just have two minor concerns left, listed below, more as ideas to improve the manuscript.

Comment 1: I think that the authors’ response to my initial major concern n°2 is really great. Maybe the authors should consider including their response to my comment in the main text of the manuscript? Right after the sentence “Both inpatient and outpatient HC-BE will be included, however, those purposively designed for autistic users will not be considered”, page 7? I copy pasted their answer below.

"Our decision to exclude care environments purposively designed for autistic users is grounded in the conceptual aims of our review. Since most autistic individuals receive care in standard HC facilities rather than specialist units, our focus is on understanding their real world experiences within care environments that were not designed with autism in mind. It is important to clarify that, to our best knowledge, national or international HC design guidance—including comprehensive frameworks such as the UK Health Building Notes—does not include autism specific accessibility requirements. This gap is also reflected in HC design guidance reviewed in other research projects across multiple countries. Consequently, any HC facility not explicitly created as an autism specific unit cannot be considered “purpose built” for autistic users. Mainstream HC settings such as general hospitals, maternity units, dental clinics, primary care facilities, and most psychiatric wards are designed for the general population, with the needs of autistic people absent from the design briefs that inform these environments. Including specialist autism designed settings would therefore introduce a fundamentally different design context and fall outside our aim of identifying BE features that affect autistic individuals in the standard HC facilities they encounter across their lifespan."

Response 1: We sincerely thank the reviewer for this excellent suggestion. We are very pleased that our previous response addressed your initial major concern. We have now incorporated the clarified explanation directly into the main text of the manuscript, immediately after the sentence: “Both inpatient and outpatient HC-BE will be included, however, those purposively designed for autistic users will not be considered” (page 7). We slightly shortened it to ensure clarity and conciseness within the flow of the manuscript:

“The decision to exclude care environments purposely designed for autistic users is grounded in the conceptual aims of this review. Since most autistic individuals receive care in standard HC facilities rather than specialist units, our focus is on understanding their real-world experiences within care environments that were not designed with autism in mind. To our best knowledge, national and international HC design guidelines do not include autism specific accessibility requirements, and this gap is reflected across design guidance reviewed in other research projects across multiple countries. As a result, most HC facilities—including general hospitals, maternity services, dental clinics, primary care, and most psychiatric wards—are not purpose‑built for autistic users. Including specialist autism designed settings would therefore introduce a fundamentally different design context and fall outside the scope of identifying BE features that affect autistic individuals in standard HC settings they encounter across their lifespan.”

Comment 2: Regarding my initial minor concern n°3, please provide a short definition or some examples of “intimate, private, and public HC-BE settings” in the main text of the manuscript, page 3, in the Introduction section.

Response 2: We thank the reviewer for this suggestion. We have carefully considered adding brief definitions of “intimate, private, and public HC‑BE settings” into the Introduction. However, the reference we cite in this section already provides the conceptual grounding needed for these terms.

Because our Introduction is structured to present the overarching theoretical framing rather than detailed typologies of HC‑BE spaces, inserting definitions for each category would disrupt the narrative flow and shift focus away from the core conceptual argument. For this reason, we have opted not to add full descriptions of all three terms directly into the main text.

Nonetheless, to support clarity, we have included a concise parenthetical clarification for “intimate settings” without lengthening the text or interrupting its trajectory: “(i.e., small, enclosed spaces with heightened vulnerability)”.

RESPONSE TO REVIEWER #5

Comment 1: The authors have addressed all five points raised in my previous review in a thorough and satisfactory manner. Specifically: (1) a Data Availability Statement has been added to the manuscript body; (2) subgroup analyses stratifying by anxiety disorder diagnosis, receipt of sensory processing therapy, age, and co-occurring conditions have been explicitly incorporated into the manuscript; (3) the Introduction now includes the prevalence of diagnosed anxiety disorders (~35%) and acknowledges that many autistic individuals experience sub-threshold anxiety symptoms as well as sensory processing differences, supported by two new references; (4) identity-first language has been applied consistently, including in Table 1 and the Discussion; and (5) Section 2.8 now explicitly names autistic individuals and their caregivers as stakeholders in dissemination. I have no further concerns. The protocol is well-structured, methodologically sound, and appropriately follows PRISMA-P guidelines. I recommend acceptance.

Response 1: Thank you very much for your positive assessment and supportive feedback. We are grateful for your careful review and are pleased that you find the protocol well‑structured, methodologically sound, and aligned with PRISMA‑P guidelines.

---

## [Editor Report · Decision Letter 3]

1 Apr 2026

Healthcare built environment and behavioural and physiological indicators of stress responses in autism spectrum disorder: Protocol for a mixed-methods systematic review

PONE-D-25-32161R3

Dear Dr. Hernandez-Garcia,

We’re pleased to inform you that your manuscript has been judged scientifically suitable for publication and will be formally accepted for publication once it meets all outstanding technical requirements.

Kind regards,

Eric J. Moody, Ph.D.

Academic Editor

PLOS One
---

## [Editor Report · Acceptance letter]

PONE-D-25-32161R3

PLOS One

Dear Dr. Hernandez-Garcia,

I'm pleased to inform you that your manuscript has been deemed suitable for publication in PLOS One. Congratulations! Your manuscript is now being handed over to our production team.

Kind regards,

on behalf of

Dr. Eric J. Moody

Academic Editor

PLOS One